# Pregestational neurological disorders among women of childbearing age—Nationwide data from a 13-year period in Hungary

Dániel Bereczki [1,2]*, Mónika Bálint[3], András Ajtay[4,5], Ferenc Oberfrank[6], Ildikó Vastagh[2,4,5]

1 János Szentágothai Doctoral School of Neurosciences, Semmelweis University, Budapest, Hungary, 2 Department of Neurology, Bajcsy-Zsilinszky Hospital and Clinics, Budapest, Hungary, 3 Centre for Economic and Regional Studies, Budapest, Hungary, 4 Department of Neurology, Semmelweis University, Budapest, Hungary, 5 MTA-SE Neuroepidemiological Research Group, ELKH, Budapest, Hungary, 6 Hungarian Academy of Sciences, Budapest, Hungary

* bereczki.daniel@hotmail.com

## Abstract

### Objectives

Comprehensive statistics evaluating pregnancies complicated by various medical conditions are desirable for the optimization of prenatal care and for improving maternal and fetal outcomes. The main objective of our study was to assess pregnancies during a 13-year study period with accompanying pregestational neurological disorders in medical history on a nationwide level.

### Methods

In the framework of the NEUROHUN 2004–2017 project utilizing medical reports submitted for reimbursement purposes to the National Health Insurance Fund, we included women with at least one labor during 2004–2016 who had at least one pregestational diagnosis of a neurological disorder received within this time frame prior to their first pregnancy during the studied period. Three-digit codes from the 10th International Classification of Diseases (ICD) were used for the identification and classification of neurological and obstetrical conditions.

### Results

Specific inclusion and exclusion criteria were employed during the study process. A total of 744 226 women have been identified with at least one delivery during the study period with 98 792 of them (13.3%) having at least one neurological diagnosis received during 2004–2016 before their first gestation in the time frame of the study. The vast majority of diagnosis codes were related to different types of headaches affecting 69 149 (9.3%) individuals. The most prevalent diagnoses following headaches were dizziness and giddiness (15 589 patients [2.1%]; nerve, nerve root and plexus disorders (10 375 patients [1.4%]); epileptic disorders (7028 patients [0.9%]); neurological diseases of vascular origin (6091 patients [0.8%]); other disorders of the nervous system (5358 patients [0.7%]); and demyelinating

**Data Availability Statement:** All relevant data are within the manuscript and its Supporting information files.

**Funding:** The work was supported by the National Brain Research Program (2017-2-1-NKP-2017-00002). There was no additional external funding received for this study. The funders had no role in study design, data collection and analysis, decision to publish, or preparation of the manuscript.

**Competing interests:** The authors have declared that no competing interests exist.

diseases of the central nervous system (2129 patients [0.3%]). The present findings of our study show high prevalence of pregestational neurological disorders, the dominance of headaches followed by the rather nonspecific diagnosis of dizziness and giddiness, the relevance of nerve, nerve root and plexus disorders and epilepsy, and the importance of cerebrovascular disorders among women of childbearing age.

## Conclusion

The present research findings can help healthcare professionals, researchers and decision makers in adopting specific health policy measures based on nationwide data and further aid the development of new diagnostic and therapeutic algorithms of various neurological manifestations concerning women of childbearing age.

## Introduction

Management of pregnancies accompanied by neurological disorders can be a complex medical challenge necessitating tight follow-up and multidisciplinary approach [1]. The relationship between gravidity and neurological conditions is bilateral [2–14]. On the one hand, physiological changes during gestation can modify the course of certain diseases [2–6]. For example, in pregnancy, migraine without aura can improve or recede in over 70% of the patients [2]. Gravidity has a disease modifying effect on multiple sclerosis (MS) with a relative reduction of relapse rates in the third semester followed by a rebound in the postpartum period [3]. In a study assessing the course of the disease in pregnancy, 50% of pregnant women with myasthenia gravis (MG) showed a deterioration (mainly during the second trimester), while 30% reported an improvement of the symptoms [4]. While the majority of epileptic women have a seizure control similar to that of the pregestational baseline, 17.3% experience an increase, while 15.9% encounter a decrease in seizure frequency [5]. Furthermore, pregnancy and puerperium present threefold higher incidence rates for ischemic and hemorrhagic strokes as compared with nonpregnant women [6]. On the other hand, neurological diseases can also have an impact on pregnancy outcomes either *per se* or indirectly, *i.e.* by their medication or via relevant diagnostic procedures affecting fetal development [7–14]. For example, patients with myasthenia gravis, epilepsy, obstructive sleep apnea or acute migraine are at increased risk of complications during gestation and/or delivery [7–10]. Compared with the general population, women suffering a stroke or a transient ischaemic attack (TIA) at fertile age show higher rates of miscarriages or fetal death throughout their lives [11]. Potential detrimental consequences of fetal exposure to certain anti-epileptic drugs, or medications used in the treatment of multiple sclerosis and myasthenia gravis have been documented [7, 12–14]. Comprehensive statistics of neurological disorders affecting future pregnancies in large-scale populations are scarce. Thus, besides concentrating on specific diseases among women in general or focusing on conditions developing during gestation itself, it seems rational to retrospectively study those women of childbearing age who have certainly become pregnant and had a delivery. It thereby is possible to evaluate the magnitude of pregnancies complicated already at the time of conception by various neurological diseases. Due to the paucity of literature, we assessed first pregnancies during a 13-year study period with accompanying pregestational neurological disorders in medical history by utilizing nationwide data on selected medical diagnoses deemed as neurological. In this context of the research gap, our aim is to optimize prenatal

care already from the very beginning of gestation by using massive data sets. In the era of big data, digital medical records can underlie such databases, particularly in countries with a single-payer state health insurance system covering the whole population [15]. In this nationwide epidemiological study, our aim was to explore and assess the first pregnancies in a 13-year period with different neurological diagnoses.

## Materials and methods

### Database design and source data for evaluation

The NEUROHUN 2004–2017 database [15] was created within the scope of the Hungarian National Brain Research Program (NBRP) from medical reports submitted for reimbursement purposes to the National Health Insurance Fund (NHIF) from all hospitals and specialist outpatient services throughout the country. Furthermore, records regarding demographic and socioeconomic factors were obtained from the Hungarian Central Statistical Office (HCSO). The full massive databank covered a 14-year period between 2004 and 2017. In the present analysis, we included women with at least one labor during 2004–2016 who had at least one pregestational diagnosis of a neurological disorder received within this time interval prior to their first pregnancy during the study period. To exclude non-clinical specialty areas (e.g. laboratory diagnostics, diagnostic imaging, physiotherapy, psychology, *etc.*), only diagnoses which had been confirmed by secondary care clinical specialties were involved in the study by the use of specific clinical specialty codes applied in Hungary. It is to be noted that primary care reports submitted by general practitioners were not included in the database. During data analysis, we used descriptive statistics. For the identification of labors and for the classification of neurological disorders, three-digit codes from the 10th International Classification of Diseases (ICD-10) [16] were applied. Conditions deemed as "neurological" were determined by the study team and comprised the following diagnostic groups (with corresponding ICD-codes):

- Malignant neoplasms of eye, brain and other parts of central nervous system (C69–C72)

- Benign neoplasm of meninges (D32)

- Benign neoplasm of brain and other parts of central nervous system (D33)

- Neoplasm of uncertain or unknown behavior of meninges (D42)

- Neoplasm of uncertain or unknown behavior of brain and central nervous system (D43)

- Diseases of the nervous system (G00-G99)

- Cerebrovascular diseases (I60-I69)

- Dizziness and giddiness (R42)

- Headache (R51)

The number of deliveries during 2004–2016 were assessed by the application of the labor-related codes O60 ("Preterm labor and delivery") and O80-O84 ("Delivery") from ICD-10 given by any clinical specialties during inpatient service. The "Delivery" group comprised the codes "Single spontaneous delivery" (O80), "Single delivery by forceps and vacuum extractor" (O81), "Single delivery by caesarean section" (O82), "Other assisted single delivery" (O83), and "Multiple delivery" (O84). Temporal distribution of the receipt of labor-related and neurological diagnoses given by clinical specialty areas enabled the identification of those patients who were diagnosed with a neurological condition during the studied years prior to their first

pregnancy in the study period as in these cases, the date of the neurological diagnosis preceded the labor-related ICD-10 code by more than nine months. Hence, those women with a sole delivery between 2004–2016 occurring in the first 9 months of the 13-year study period were excluded from the present analysis. During the retrospective study of medical records, centrally anonymized data were provided by the National Health Insurance Fund. By the use of encrypted codes derived from original patient identifiers, record linkage was also possible. Study approval was provided by the Ethics Committee of Semmelweis University, Budapest, Hungary (Approval No.: SE TUKEB 88-1/2015) and data management was in line with personal data protection rules. Primary data acquisition was performed by a research assistant with an IT specialization and extensive experience in studying medical records of patients. For the final analysis, results of individual searches in the database were exported to excel files used for further evaluation during the final analysis.

## Results

### a) The general prevalence of pregestational neurological disorders

By the use of the abovementioned inclusion and exclusion criteria, 744 226 women were identified with at least one delivery during the study period. Of those having at least one neurological diagnosis received during the 13-year time frame before their first pregnancy in the studied interval resulted in 98 792 cases. Thus, 13.3% of abovementioned 744 226 women had already received at least one neurological diagnosis during 2004–2016 prior to their first gestation in the study period and became pregnant with that in mind. Table 1 shows the number of patients classified by neurological diagnoses received during the 13-year time frame before their first pregnancy within the study interval. As the majority of these ICD-10 codes represent diagnostic groups, patients receiving more than one ICD-10 code within one diagnostic group were counted only once (the number of all diagnoses without filtering repetitions are presented in parentheses). Nevertheless, individual women could appear in multiple different diagnostic categories. Table 2 shows detailed data on patient numbers for "Episodic and paroxysmal disorders" (G40-G47).

### b) The prevalence of specific pregestational neurological disorders

(i) Headaches (G43-G44; R51).   Analysis of the data demonstrated a massive dominance of diagnoses referring to different types of headaches. As individual women could appear in more types of headache categories adding up a total of 85 091 diagnoses, such overlaps were excluded resulting in 69 149 patients. The group contains "Migraine" (G43; 12 909 cases [18.7% of headache patients]), "Other headache syndromes" (G44; 21 086 cases [30.5% of headache patients], and "Headache" (R51; 51 096 cases [73.9% of headache patients]). Notably, the abovementioned 69 149 individuals displayed 70% of women with at least one neurological diagnosis received between 2004 and 2016 prior to their first pregnancy during the studied years, and 9.3% of all women with at least one labor in the study period.

(ii) Dizziness and giddiness (R42).   In terms of prevalence, different types of headaches were followed by the diagnosis of "Dizziness and giddiness" (R42) affecting 15 589 cases, thereby making this category the second most common pregestational neurological diagnosis with 15.8%. This code comprised 2.1% of all individuals with at least one delivery during the study period.

(iii) Nerve, nerve root and plexus disorders (G50-G59).   "Nerve, nerve root and plexus disorders" (G50-G59; 10 375 patients) ranked third among the top ICD-10 categories affecting 10.5% of women with at least one neurological diagnosis received during 2004–2016 before their first pregnancy in the study period. This group involved 1.4% of all women with at least

**Table 1. Classification of patients by neurological diagnoses received in 2004–2016 prior to first pregnancy during the study period.**

| ICD-10 numerical code | Diagnostic classification | Number of patients (Number of diagnoses, if applicable) |
|---|---|---|
| C69-C72 | Malignant neoplasms of eye, brain and other parts of central nervous system | 256 (263) |
| D32-D33 | Benign neoplasm of meninges (D32) | 721 (756) |
| | Benign neoplasm of brain and other parts of central nervous system (D33) | |
| D42-D43 | Neoplasm of uncertain or unknown behaviour of meninges (D42) | 257 (259) |
| | Neoplasm of uncertain or unknown behaviour of brain and central nervous system (D43) | |
| G00-G09 | Inflammatory diseases of the central nervous system | 423 (481) |
| G10-G14 | Systemic atrophies primarily affecting the central nervous system | 66 (67) |
| G20-G26 | Extrapyramidal and movement disorders | 1551 (1606) |
| G30-G32 | Other degenerative diseases of the nervous system | 180 (180) |
| G35-G37 | Demyelinating diseases of the central nervous system | 2129 (2381) |
| G40-G47 | Episodic and paroxysmal disorders | 40814 (46712) |
| G50-G59 | Nerve, nerve root and plexus disorders | 10375 (10877) |
| G60-G64 | Polyneuropathies and other disorders of the peripheral nervous system | 1107 (1194) |
| G70-G73 | Diseases of myoneural junction and muscle | 655 (722) |
| G80-G83 | Cerebral palsy and other paralytic syndromes | 738 (872) |
| G90-G99 | Other disorders of the nervous system | 5358 (5598) |
| I60-I69 | Cerebrovascular diseases | 3082 (3669) |
| R42 | Dizziness and giddiness | 15589 |
| R51 | Headache | 51096 |

one labor during the studied years. Within this diagnostic interval, "Mononeuropathies of upper limb" (G56) were the most represented disorders with 28.5%, affecting 2953 cases. Other relevant subgroups were "Disorders of trigeminal nerve" (G50) with 19.4%, involving 2013 patients, "Nerve root and plexus disorders" (G54) with 19.3%, comprising 2006 patients, "Facial nerve disorders" (G51) with 17.5%, concerning 1818 cases and "Other mononeuropathies" (G58) with 11.2%, affecting a total of 1161 cases in the data sets.

**Table 2. Number of patients with G40-G47 diagnoses received in 2004–2016 prior to first pregnancy during the study period.**

| ICD-10 numerical code | Diagnostic classification | Number of patients (Number of diagnoses, if applicable) |
|---|---|---|
| G40-G41 | Epilepsy (G40) | 7028 (7089) |
| | Status epilepticus (G41) | |
| G43 | Migraine | 12909 |
| G44 | Other headache syndromes | 21086 |
| G45 | Transient cerebral ischaemic attacks and related syndromes | 3508 |
| G46 | Vascular syndromes of brain in cerebrovascular diseases | 63 |
| G47 | Sleep disorders | 2057 |

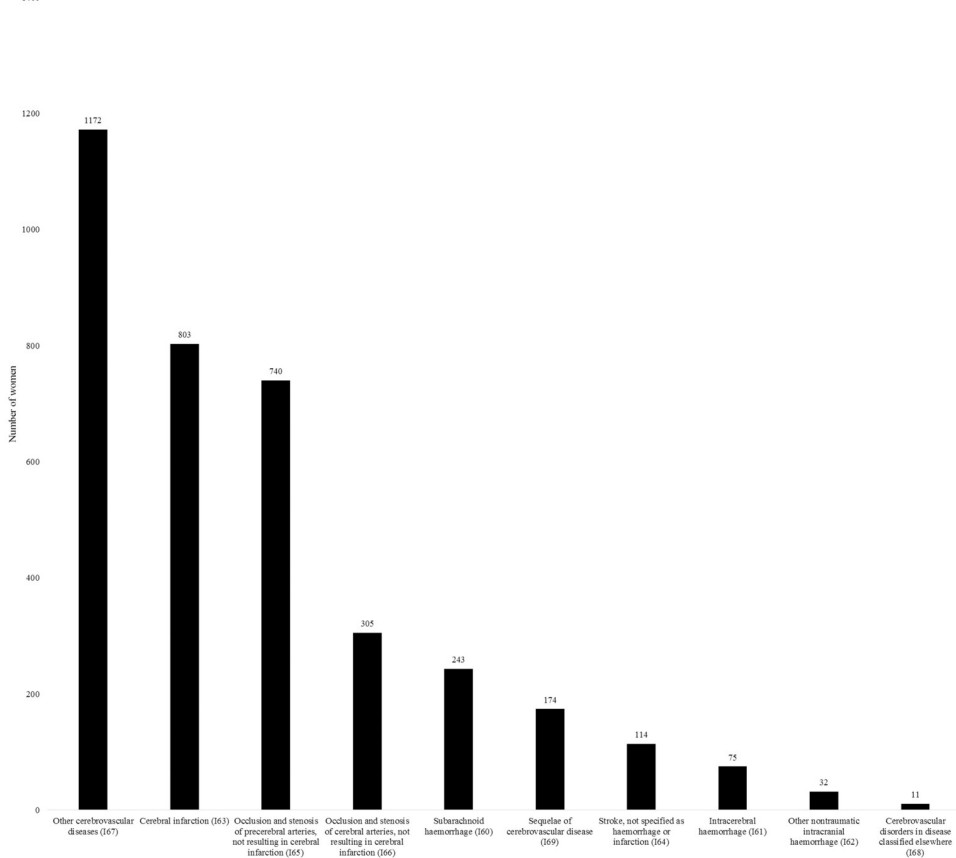

**Fig 1. Diagnoses of "Cerebrovascular diseases" (I-numerical codes) received during 2004–2016 prior to first pregnancy within the study period.**

**(iv) Epilepsy and status epilepticus (G40-G41).** The G40-G41 diagnostic group including "Epilepsy" (G40) and "Status epilepticus" (G41) involved 7028 women, comprising 7.1% of those with a pregestational neurological diagnosis prior to their first gestation during the study interval. Of all women with at least one labor during 2004–2016, 0.9% received the diagnosis of epilepsy and/or status epilepticus during this time period prior to their first pregnancy within the studied years.

**(v) Cerebrovascular disorders (G45-G46; I60-I69).** Although the ICD-10 system classifies the category "Transient cerebral ischaemic attacks and related syndromes" (G45) and "Vascular syndromes of brain in cerebrovascular diseases" (G46) separately, it seems reasonable to handle them along with "Cerebrovascular diseases" (I60-I69) when assessing neurological diseases of vascular origin. These 6091 patients displayed 6.2% of women with at least one labor and a pregestational neurological diagnosis prior to their first pregnancy between 2004 and 2016 and represented 0.8% of all individuals with at least one delivery during the studied time frame, making cerebrovascular disorders almost as prevalent as epilepsy among these women of childbearing age. Within the category of cerebrovascular disorders, "Transient cerebral ischaemic attacks and related syndromes" were the most represented (3508 cases), followed by diseases of the I60-I69 group displayed in Fig 1. The main representatives of this latter ICD-10 group were "Other cerebrovascular diseases" (I67; 1172 cases), "Cerebral

infarction" (I63; 803 cases) and "Occlusion and stenosis of precerebral arteries, not resulting in cerebral infarction" (I65; 740 patients).

 **(vi) Other disorders of the nervous system (G90-G99).** This ICD-10 category containing mainly unspecific neurological disorders ranked among the most prevalent categories (5358 patients; 5598 diagnoses) owing to its subgroups "Disorders of autonomic nervous system" (G90) with 51.5%, affecting 2762 cases and "Other disorders of brain" (G93) with 37.9%, including 2031 cases.

 **(vii) Demyelinating disorders of the central nervous system (G35-G37).** This diagnostic interval of 2129 patients (2381 diagnoses) was mostly represented by "Multiple sclerosis" (G35) with 82.2%, affecting 1751 individuals and thereby displaying a prevalence of 0.2% among all women with at least one delivery during the study time frame.

 **(viii) Other neurological disorders.** In terms of prevalence, the beforementioned ICD-10 categories were followed by "Sleep disorders" (G47; 2057 cases); "Extrapyramidal and movement disorders" (G20-G26; 1551 cases) and "Polyneuropathies and other disorders of the peripheral nervous system" (G60-G64; 1107 cases). Further diagnostic groups comprising less women are presented in Table 1.

## Discussion

In the present study, our objective was to assess the magnitude of first pregnancies of a 13-year study period with previous neurological diagnoses received during the studied years on a nationwide level, by deriving data from the NEUROHUN 2004–2017 project. The NEURO-HUN database utilizing medical reports submitted for reimbursement purposes from a single-payer health insurance system covering the whole population has proven valuable in characterizing epidemiological features of various neurological conditions, such as ischaemic stroke, headache, Parkinson's disease or multiple sclerosis [17–21].

### a) Pregestational neurological disorders in general

After the application of the above detailed specific inclusion and exclusion criteria, it has been revealed that out of the 744 226 studied women, 98 792 (13.3%) had their first pregnancy within the given time interval already possessing at least one pregestational neurological diagnosis received during the study period. This can be considered a surprisingly high prevalence especially for such a narrowly-defined population.

### b) Specific neurological disorders before conception

 **(i) Headaches.** The vast majority of "neurological" ICD-10 codes were related to different types of headaches, with 85 091 diagnoses affecting 69 149 individuals resulting in a prevalence of 70% among the studied women having at least one neurological diagnosis received prior to their first pregnancy during 2004–2016 and affecting 9.3% of those with at least one delivery in the study period. The high volume of these diagnoses was in correspondence with the results of the Global Burden of Disease (GBD) Study 2016 with tension-type headache and migraine ranking among the top ten causes with the greatest prevalence worldwide [22]. Notably, headaches are most burdensome in women of childbearing age (between ages 15 and 49 years) making 11.2% of all years of life lived with disability (YLD) in this age group and sex [23]. Also, according to the latest GBD Study (2019), migraine remained the top cause of YLD among young women and took second place in all age groups [24]. When looking at disability-adjusted life years (DALYs), migraine ranked first among young adult women [24]. It should be taken into account, especially when comparing the abovementioned numbers to previous data showing a global prevalence of current headache being 47% [25], that prevalence

values of this study are based on medical reports representing only individuals seeking specialist medical care in response to their symptoms with records submitted by general practitioners being not included. It is also to be noted that among the 69 149 headache patients, 73.9% received at least the nonspecific R51 code "Headache". This could partially be attributed to the fact that, as per study design, neurological diagnoses could have been given by any kind of clinical specialties, with the possibility that non-neurologists tend to make no slight distinctions between different headache types during documentation.

**(ii) Dizziness and giddiness.** In terms of prevalence, although far behind headaches, the rather nonspecific diagnosis of "Dizziness and giddiness" (R42) ranked second among the most common neurological diagnoses in medical history with 15 589 patients. According to internal medicine outpatient service data, among leading complaints, dizziness is the third most common general symptom [26]. There are estimations regarding the lifetime prevalence being 15–35% in the general population [27]. According to recent systematic review data based on primary care consultations, the two most common reasons for dizziness were of cardiovascular and peripheral otologic origin [28]. However, it is to be noted, that besides otologic/vestibular and cardiovascular origins, the etiological spectrum of dizziness is considerably wide, including also following diagnostic groups: respiratory, neurologic (including cerebrovascular), metabolic, injury/poisoning, psychiatric, digestive, genitourinary, and infectious [29]. Furthermore, assessments regarding prevalence need to be evaluated cautiously as several different types of complaints can be described as "dizziness" by the patient (e.g. vertigo, disequilibrium, faintness, visual and gait disturbances, anxiety). When interpreting the relatively low prevalence in the current study with 2.1% of women with at least one delivery during the study period being affected, it has to be taken into account that more specific diagnoses underlying this general complaint (e.g. benign paroxysmal vertigo, syncope and collapse, abnormalities of gait and mobility, diplopia, phobic anxiety disorders) were not included in this category.

**(iii) Nerve, nerve root and plexus disorders.** Within the third most common pregestational neurological diagnostic category of "Nerve, nerve root and plexus disorders" (G50-G59) affecting 10 375 patients, "Mononeuropathies of the upper limb" (G56) were the most represented involving lesions of the median, ulnar and radial nerves. With an estimated lifetime risk of 10%, affecting women more likely than men and having an increasing incidence with age, carpal tunnel syndrome is the most common focal, compressive neuropathy of the upper extremity, followed by ulnar neuropathy due to entrapment in the elbow region [30–33]. Radial neuropathies generally can be caused either by external nerve compression (Saturday night palsy) or by trauma usually associated with fracture of the humerus [30, 32]. Mononeuropathies of the upper limb were followed by "Disorders of trigeminal nerve" (G50), representing 19.4% of nerve, nerve root and plexus disorders. It is to be noted that oro-facial pain has a high prevalence among women (30%) with the age group 18–25 years being the most affected [34]. The ICD-10 subgroup G50 involves atypical facial pain, an underdiagnosed, debilitating condition with poor prognosis affecting most likely women in their forties [35]. Being in this subgroup, trigeminal neuralgia also has the highest prevalence among women older than 40 years [36]; however, it may also need to be taken into account that multiple sclerosis being the most prevalent in the sex and age group of our study population [37] is associated with a 20-fold higher prevalence of trigeminal neuralgia [38]. Closely following disorders of the trigeminal nerve, "Nerve root and plexus disorders" (G54) displayed 19.3% of diagnoses within this diagnostic interval (G50-G59). This could be explained by the fact, that these conditions can be etiologically related to neck pain and low back pain, the two main causes of disability of musculoskeletal origin [22]. These complaints ranked globally as the fourth leading cause of disability-adjusted life years (DALY) following ischemic heart disease, cerebrovascular

disease, and lower respiratory infection [39]. According to the GBD Study 2019, low back pain remained the leading cause of age-standardized YLD worldwide with a higher burden in women [40]. "Facial nerve disorders" (G51) ranked fourth among nerve, nerve root and plexus disorders with 17.5% of the diagnoses. Most well-known condition of this category is Bell's palsy, displaying the highest incidence between the ages of 15 and 45 years and having a complete recovery in 80% of non-pregnant women in this age group [41]. Notably, peripheral facial nerve palsy occurring during gestation has a much worse prognosis with only 61% of patients recovering fully [41]. Affecting 11.2% of patients within abovementioned diagnostic interval (G50-G59), "Other mononeuropathies" (G58) including e.g. intercostal neuropathy and mononeuritis multiplex, was the fifth most prevalent diagnosis among nerve, nerve root and plexus disorders.

**(iv) Epileptic disorders.** Ranking after headaches, dizziness and giddiness, and nerve, nerve root and plexus disorders, epileptic disorders involving "Epilepsy" (G40) and "Status epilepticus" (G41) involved 7 028 patients of the study population. Although data are partly conflicting in this regard, it seems that in comparison to women without epilepsy, women with epilepsy over 25 years of age have significantly lower birth rates [42]. This fact may have several causes, partially attributed to the potential side effects of the antiepileptic treatment (influence on hormone levels and thereby sexual function; fear of fetal malformations) but psychosocial factors, psychiatric comorbidities or the fear of seizures during pregnancy may also take part in the above observation [42]. Nevertheless, according to recent data, women with epilepsy seeking pregnancy have a similar likelihood of getting pregnant and similar live birthrates when compared to women without epilepsy [43]. The prevalence of the diagnosis of epileptic disorders being 0.9% among women with at least one delivery in the given time period corresponds to the data on epilepsy affecting almost 1% of the population [44].

**(v) Cerebrovascular disorders.** According to literature data, 16 to 59 per 100 000 women of childbearing age are affected by stroke per year [11, 45]. In young women, menstruation, pregnancy and the early postpartum period pose a challenge in patient care in case of acute ischemic stroke. According to the latest European Stroke Organisation (ESO) guidelines, although available data do not allow evidence-based recommendations, expert consensus statements rather suggest active treatment (intravenous thrombolysis and/or mechanical thrombectomy) in selected cases [46]. Surprisingly, neurological diseases of vascular origin affecting 6091 women in our study population ranked among the most prevalent neurological disorders. The prevalence of 0.8% among all women having at least one labor during the study period points out the relevance of cerebrovascular disorders among women of childbearing age. This high prevalence can be attributed to the followings: (1) Age is a well-known non-modifiable cardiovascular risk factor [47]. During the last decades, especially in developed countries, there has been a dramatic increase in the number of women giving birth at an advanced age. In the United States, between 2007 and 2016, births rates have risen 11% for women in their late thirties and 19% for women in their early forties [48]. Also, 20% of babies born in England and Wales in 2013 had mothers aged 35 and over at the time of birth [49]. With advancing age, the increasing prevalence of different chronic conditions should also need to be taken into account. (2) The diagnoses of cerebrovascular disorders were given by any clinical specialists, including non-neurologists, with the possibility of assessing many–especially transient–symptoms as being of vascular origin. The latter hypothesis is supported by the relatively high prevalence of "Transient cerebral ischaemic attacks and related disorders" (G45) and by the rather aspecific group of "Other cerebrovascular diseases" (I67) being the most prevalent category within the I60-I69 interval. Nevertheless, it should be noted that the latter category includes diagnoses such as nonruptured cerebral aneurysm, cerebral atherosclerosis, progressive vascular leukoencephalopathy, hypertensive encephalopathy or

nonpyogenic sinus vein thrombosis which could also contribute to the high prevalence of this diagnostic category.

**(vi) Other disorders of the nervous system.** The ICD-10 interval G90-G99 representing "Other disorders of the nervous system" included 5358 patients with two major subcategories displaying almost 90% of the diagnoses: "Disorders of autonomic nervous system" (G90–51.5%) and "Other disorders of brain" (G93–37.9%). The dominance of G90, an ICD-10 code rather rarely given by neurologists, may be attributed to the fact that diagnoses received from non-neurologists were also included in the study with the potential tendency to use this diagnostic category for more general complaints, e.g. dizziness, fainting, orthostatic hypotension, diarrhea, urinary incontinence or vaginal dryness. The high prevalence of subgroup G93 can be explained by the involvement of conditions such as cerebral cysts, benign intracranial hypertension, unspecified encephalopathy, cerebral edema, and the categories of other specified and other unspecified disorders of the brain.

**(vii) Demyelinating diseases of the central nervous system.** Multiple sclerosis is one of the most common causes of neurological disability in young people with the onset of the disease peaking between 20 and 40 years of age and with women being 2–3 times more frequently affected than men [37]. Epidemiological studies conducted in Csongrád county in Hungary utilizing data from a local MS register resulted in an increasing crude MS prevalence in females from 128.6 per 100 000 in 2013 to 149.3 per 100 000 in 2019 [50, 51]. A recent study deriving data from healthcare administrative records showed MS being more prevalent in Hungary than previously thought with the same tendency in numbers increasing from 150.8 per 100 000 to 179.5 per 100 000 between 2010 and 2015 among women [21]. Further statistics from Central Europe show a crude prevalence of MS up to 240 per 100 000 among females [52, 53]. According to our study, over 80% of women (1751 patients) within the category of "Demyelinating diseases of the central nervous system" (G35-G37) received the diagnosis "Multiple sclerosis" (G35). The displayed relatively high prevalence of 0.2% among all women with at least one labor during the study period could be attributed to the gender and age characteristics of the study population (i.e. women of childbearing age) and to the utilization of healthcare administrative data provided by all clinical specialties.

## Strengths and limitations

Strengths of our study can be highlighted in the followings: (1) the analysis of reports from a single-payer state health insurance system enabling the full coverage of national data; (2) long timeframe covering a time period from 2004 to 2016; (3) the above points resulted in a substantial number of cases; (4) such comprehensive data in a medical frontier are scarce and require further investigation. Also, our study had a few limitations: (1) the definition of "neurological" disorders beyond the G-category ("Diseases of the nervous system") of ICD-10 has been determined by the authors; (2) neurological diagnoses were given by all clinical specialties without being necessarily confirmed by neurologists; (3) the dataset contained diagnoses given by secondary care specialists, however medical reports submitted by general practitioners were not involved in the study; (4) evaluation of the diagnoses was restricted by the use of 3-digit ICD-10 codes not allowing further, more sophisticated assessment; (5) having a defined time interval, only neurological diagnoses received within the given period were taken into account–also, the above principle was applied when assessing first pregnancies during the study period.

## Conclusions

As the scientific literature is scarce on data from large populations concerning the wide range of neurological disorders, nationwide statistics enabling the optimization of prenatal care are

desirable. As estimations on disease prevalence deriving data extracted from healthcare administrative reports have proven useful before, we applied this method in our study to gather information about neurological diagnoses received prior to first pregnancies during a specific 13-year interval. The main findings of our study were the high prevalence of pregestational neurological disorders, the dominance of headaches followed by the rather nonspecific diagnosis of dizziness and giddiness, the relevance of nerve, nerve root and plexus disorders and epilepsy, and the importance of cerebrovascular disorders among women of childbearing age. Further studies on specific disease categories including pregnancy outcomes, or neurological confirmation of the diagnoses would help us refine the characterization of this specific population.

## Supporting information

**S1 File.**
(XLSX)

## Acknowledgments

We thank Prof. Dr. Daniel Bereczki (Semmelweis University) for critically reviewing the manuscript and for providing valuable feedback.

## Author Contributions

**Conceptualization:** Dániel Bereczki, András Ajtay, Ildikó Vastagh.

**Data curation:** Dániel Bereczki, Mónika Bálint, András Ajtay, Ferenc Oberfrank, Ildikó Vastagh.

**Formal analysis:** Dániel Bereczki, Mónika Bálint, Ildikó Vastagh.

**Investigation:** Dániel Bereczki, András Ajtay.

**Methodology:** Dániel Bereczki, Mónika Bálint, András Ajtay, Ildikó Vastagh.

**Resources:** András Ajtay, Ferenc Oberfrank.

**Software:** Mónika Bálint.

**Supervision:** Ildikó Vastagh.

**Validation:** András Ajtay, Ferenc Oberfrank, Ildikó Vastagh.

**Writing – original draft:** Dániel Bereczki.

**Writing – review & editing:** Dániel Bereczki, Ildikó Vastagh.

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
