## [Editor Report · Decision Letter 0]

12 May 2021

PONE-D-21-09199

Pregestational neurological disorders among women of childbearing age - nationwide data from a 13-year period in Hungary

PLOS ONE

Dear Dr. Bereczki,

Thank you for submitting your manuscript to PLOS ONE. After careful consideration, we feel that it has merit but does not fully meet PLOS ONE’s publication criteria as it currently stands. Therefore, we invite you to submit a revised version of the manuscript that addresses the points raised during the review process.

We look forward to receiving your revised manuscript.

Kind regards,

Basvarajaiah D. M., ph.D

Academic Editor

PLOS ONE

Additional Editor Comments:

Below are a few comments that the authors can consider.

(i)The author unable to describe the research gap and rationality is not up to the mark, the practical utility of the research is well planned and derived properly. In Methodological section, I have not seen , what are the tests used for assessing the patients and selection of concomitant variables.

(ii) In result part, the research hypothesis is not tested by relevant statistical methods .The flow of resulted part is not fulfilled our journal criteria. Plz repharse the sentence and describe the result part by using accurate estimation of variables to be tested by statistical methods.

(iii) Discussion and conclusion part is not fulfilled our journal criteria-Major revision should be necessary by the author

Decision of the research paper: Major revision and resubmission, because topic is more useful for the scientific community

Requested the Author, plz Rephrase the above limitation and comments.

Journal Requirements:

For additional information about PLOS ONE ethical requirements for human subjects research, please refer to " ext-link-type="uri" xlink:type="simple">http://journals.plos.org/plosone/s/submission-guidelines#loc-human-subjects-research."

 [The work was partly supported by the National Brain Research Program (2017-2-1-NKP-2017-00002). The funders had no role in study design, data collection and analysis, decision to publish, or preparation of the manuscript.]. 
---

## [Author Response · Author response to Decision Letter 0]

13 Jun 2021

Dear Editor,

thank you for the valuable feedback provided on our manuscript Bereczki D Jr. et al: Pregestational neurological disorders among women of childbearing age – nationwide data from a 13-year period in Hungary submitted with the intention to publish it in the journal PLOS ONE.

Please, find our responses to the reviewers’ comments:

“(i)The author unable to describe the research gap and rationality is not up to the mark, the practical utility of the research is well planned and derived properly. In Methodological section, I have not seen , what are the tests used for assessing the patients and selection of concomitant variables.”

The relevant part of the manuscript in the “Introduction” part on describing the research gap and rationality has been updated as follows: 

“Comprehensive statistics from large population on neurological disorders affecting future pregnancies are scarce, thus besides concentrating on specific diseases among women in general or focusing on conditions developing during gestation itself, it seems rational to retrospectively study those women of childbearing age who have certainly become pregnant and had a delivery and thereby evaluate the magnitude of pregnancies complicated already at the time of conception by various neurological diseases. Nationwide data on such selected population would be desirable to further optimize prenatal care from the very beginning of gestation and thereby improve maternal and fetal outcomes.”

The “Materials and methods” section of the manuscript has been divided into two main parts: “Database design” and “Data evaluation”. In the latter part, it is now emphasized that we have used descriptive statistics by inserting the following sentence to the very beginning of the section:

“During data analysis, we used descriptive statistics.”

Also, for the sake of clarity, we updated the wording in two sentences:

1. Instead of “For the identification and classification of neurological and obstetrical conditions, three-digit codes from the 10th International Classification of Diseases (ICD-10) were used.”, the updating wording is as follows: “For the identification of labours and for the classification of neurological disorders, three-digit codes from the 10th International Classification of Diseases (ICD-10) were applied.”

2. Instead of “Temporal distribution of obstetrical and neurological diagnoses enabled the identification (…)”, the updated wording is as follows: “Temporal distribution of the receipt of labor-related and neurological diagnoses given by clinical specialty areas enabled the identification (…)”.

Also, in the “Materials and methods” section, the ethics statement has been updated emphasizing that this was a retrospective study of medical records and also approval number provided by the Ethics Committee of Semmelweis University, Budapest, Hungary was added. The updated wording is as follows: “During the retrospective study of medical records, data anonymization was provided by the use of encrypted codes derived from original patient identifiers, thereby also making record linkage possible. Study approval was provided by the Ethics Committee of Semmelweis University, Budapest, Hungary (Approval No.: SE TUKEB 88-1/2015) and data management was in line with personal data protection rules.”

“(ii) In result part, the research hypothesis is not tested by relevant statistical methods .The flow of resulted part is not fulfilled our journal criteria. Plz repharse the sentence and describe the result part by using accurate estimation of variables to be tested by statistical methods.”

As pointed out above and also emphasized in the revised manuscript, we used descriptive (and not comparative) statistics during our work. As per SAMPL guidelines, we updated the “Results” part by providing numerators and denominators to all percentages.

“(iii) Discussion and conclusion part is not fulfilled our journal criteria”

The “Discussion” part has been split into three subunits with following titles: “Pregestational neurological disorders in general”; “Specific neurological disorders before conception”; “Strengths and limitations”. The subunit “Specific neurological disorders before conception” has been further divided into subsections.

The “Conclusion” part has been reworded putting more emphasis on the research gap. The updated wording is as follows: “As the scientific literature is scarce on data from large populations concerning the wide range of neurological disorders, nationwide statistics enabling the optimization of prenatal care are desirable.”

As per request, funding statement has been amended and included in the updated cover letter as follows: “The work was supported by the National Brain Research Program (2017-2-1-NKP-2017-00002). There was no additional external funding received for this study. The funders had no role in study design, data collection and analysis, decision to publish, or preparation of the manuscript.” 

We hope that after considering our revisions and comments, you will find our manuscript worth to consider it for publication.

Sincerely yours,

Dániel Bereczki Jr., MD

Corresponding author

E-mail: bereczki.daniel@hotmail.com

---

## [Decision Letter · Decision Letter 1]

30 Jun 2022

PONE-D-21-09199R1Pregestational neurological disorders among women of childbearing age - nationwide data from a 13-year period in HungaryPLOS ONE

Dear Dr. Bereczki,

Thank you for submitting your manuscript to PLOS ONE. After careful consideration, we feel that it has merit but does not fully meet PLOS ONE’s publication criteria as it currently stands. Therefore, we invite you to submit a revised version of the manuscript that addresses the points raised during the review process.

The authors provide their calculations for percentages, which doesn’t seem completely necessary (at least the “x 100”) piece, and the “Pregestational neurological disorders in general” piece of the discussion is a little redundant with the methods section.  The English could be more idiomatic (for instance, the section added in the introduction has several long and wordy sentences) but doesn’t seem to be incorrect.

We look forward to receiving your revised manuscript.

Kind regards,

Lucinda Shen

Staff Editor

on behalf of 

Emily W. Harville

Academic Editor

PLOS ONEo:p/o:p

Journal Requirements:

Reviewers' comments:

Reviewer's Responses to Questions

**Comments to the Author**

1. If the authors have adequately addressed your comments raised in a previous round of review and you feel that this manuscript is now acceptable for publication, you may indicate that here to bypass the “Comments to the Author” section, enter your conflict of interest statement in the “Confidential to Editor” section, and submit your "Accept" recommendation.

Reviewer #1: All comments have been addressed

2. Is the manuscript technically sound, and do the data support the conclusions?

Reviewer #1: Yes

3. Has the statistical analysis been performed appropriately and rigorously? 

Reviewer #1: I Don't Know

4. Have the authors made all data underlying the findings in their manuscript fully available?

Reviewer #1: Yes

5. Is the manuscript presented in an intelligible fashion and written in standard English?

Reviewer #1: Yes

6. Review Comments to the Author

Reviewer #1: The most important practical observation is that neurological diagnoses are not just from neurologists - so incidence data can be misleading. I feel that it is a significant shortcoming that there are no data on the course and outcome of pregnancies. This would significantly increase the value of the article.

7. PLOS authors have the option to publish the peer review history of their article (what does this mean?). If published, this will include your full peer review and any attached files.

Reviewer #1: No

---

## [Author Response · Author response to Decision Letter 1]

24 Aug 2022

Dear Editor,

thank you for the valuable feedback provided on our manuscript Bereczki D Jr. et al: Pregestational neurological disorders among women of childbearing age – nationwide data from a 13-year period in Hungary submitted with the intention to publish it in the journal PLOS ONE.

Please, find our responses to the reviewers’ comments:

(i) „The authors provide their calculations for percentages, which doesn’t seem completely necessary (at least the “x 100”) piece.”

The concerned calculations have been removed from the revised manuscript.

(ii) “The “Pregestational neurological disorders in general” piece of the discussion is a little redundant with the methods section.” 

In order to dissolve the abovementioned redundancy, the concerned part of the Discussion section has been revised and simplified accordingly. The updated wording is as follows: “After the application of the above detailed specific inclusion and exclusion criteria, it has been revealed that (…)”

(iii) “The English could be more idiomatic (for instance, the section added in the introduction has several long and wordy sentences) but doesn’t seem to be incorrect.”

The concerned section added in the Introduction part has been split into three shorter sentences. The updated version is as follows: “Comprehensive statistics of neurological disorders affecting future pregnancies in large-scale populations are scarce. Thus, besides concentrating on specific diseases among women in general or focusing on conditions developing during gestation itself, it seems rational to retrospectively study those women of childbearing age who have certainly become pregnant and had a delivery. Thereby it is possible to evaluate the magnitude of pregnancies complicated already at the time of conception by various neurological diseases.”

(iv) As per request, additional literature citations have been added to the manuscript. 

The newly added references (Reference 1, 24, 28, 40 and 46) are the following:

• Bereczki D Jr. Terhesség és akut ischaemiás stroke [Pregnancy and acute ischemic stroke]. Orv Hetil 2016;157:763-766.

• Steiner TJ, Stovner LJ, Jensen R, Uluduz D, Katsarava Z. Migraine remains second among the world's causes of disability, and first among young women: findings from GBD2019. J Headache Pain 2020;21:137.

• Bösner S, Schwarm S, Grevenrath P, Schmidt L, Hörner K, Beidatsch D, et al. Prevalence, aetiologies and prognosis of the symptom dizziness in primary care – a systematic review. BMC Fam Pract 2018;19:33.

• Chen S, Chen M, Wu X, Lin S, Tao C, Cao H, et al. Global, regional and national burden of low back pain 1990-2019: a systematic analysis of the Global Burden of Disease study 2019. J Ortop Translat 2021;32:49-58.

• Kremer C, Gdovinova Z, Bejot Y, Heldner MR, Zuurbier S, Walter S, et al. European Stroke Organisation guidelines on stroke in women: Management of menopause, pregnancy and postpartum. Eur Stroke J 2022;7:I-XIX.

We hope that after considering our revisions and comments, you will find our manuscript worth to consider it for publication.

Sincerely yours,

Dániel Bereczki Jr., MD

Corresponding author

---

## [Editor Report · Decision Letter 2]

1 Sep 2022

PONE-D-21-09199R2Pregestational neurological disorders among women of childbearing age - nationwide data from a 13-year period in HungaryPLOS ONE

Dear Dr. Bereczki,

Thank you for submitting your manuscript to PLOS ONE. After careful consideration, we feel that it has merit but does not fully meet PLOS ONE’s publication criteria as it currently stands. Therefore, we invite you to submit a revised version of the manuscript that addresses the points raised during the review process.

The substantive issues have been corrected, so I had just a few more suggestions to make the English more idiomatic.==============================

We look forward to receiving your revised manuscript.

Kind regards,

Emily W. Harville

Academic Editor

PLOS ONE

Journal Requirements:

1. We suggest you thoroughly copyedit your manuscript for language usage, spelling, and grammar. If you do not know anyone who can help you do this, you may wish to consider employing a professional scientific editing service. 

Additional Editor Comments:

The authors have addressed the substantive comments and I have just a few English suggestions.

Throughout - clarify that GP is not considered a specialty; diagnoses were taken from secondary care practitioners. (to my understanding.)

Objectives: “The” main objective

Remove “massive”

Conclusion: “The” present research, algorithms

Line 82 It thereby rather than Thereby it

105 remove comma after specialties

106 remove comma after noted

128 remove comma after patients

138 an IT specialization and extensive

145 change have been to were

146 change Out of them to Of those

185 move first clause to end of sentence

278 headache add “sufferers” or “patients”

279 move comma after that

280 put comma after specialties

322 40 years [36], change to semicolon

335 comma after palsy

444 add “The” main
---

## [Author Response · Author response to Decision Letter 2]

4 Sep 2022

Dear Editor,

thank you for the additional comments provided on our manuscript Bereczki D Jr. et al: Pregestational neurological disorders among women of childbearing age – nationwide data from a 13-year period in Hungary submitted with the intention to publish it in the journal PLOS ONE.

Please, find our responses to the comments:

(i) “Throughout - clarify that GP is not considered a specialty; diagnoses were taken from secondary care practitioners. (to my understanding.)”

 For the sake of clarity, we reworded the concerned parts of the manuscript:

In the Materials and methods section, the updated wording is as follows: “To exclude non-clinical specialty areas (e.g. laboratory diagnostics, diagnostic imaging, physiotherapy, psychology, etc.), only diagnoses which had been confirmed by secondary care clinical specialties were involved in the study by the use of specific clinical specialty codes applied in Hungary. It is to be noted that primary care reports submitted by general practitioners were not included in the database.”

In the Strengths and limitations section, the updated wording is as follows: “(3) the dataset contained diagnoses given by secondary care specialists, however medical reports submitted by general practitioners were not involved in the study”

(ii) All of the grammatical suggestions have been implemented in the updated manuscript.

We hope that after considering our revisions and comments, you will find our manuscript worth to consider it for publication.

Sincerely yours,

Dániel Bereczki Jr., MD

Corresponding author

---

## [Editor Report · Decision Letter 3]

7 Sep 2022

Pregestational neurological disorders among women of childbearing age - nationwide data from a 13-year period in Hungary

PONE-D-21-09199R3

Dear Dr. Bereczki,

We’re pleased to inform you that your manuscript has been judged scientifically suitable for publication and will be formally accepted for publication once it meets all outstanding technical requirements.

Kind regards,

Emily W. Harville

Academic Editor

PLOS ONE
---

## [Editor Report · Acceptance letter]

12 Sep 2022

PONE-D-21-09199R3 

Pregestational neurological disorders among women of childbearing age – nationwide data from a 13-year period in Hungary 

Dear Dr. Bereczki:

I'm pleased to inform you that your manuscript has been deemed suitable for publication in PLOS ONE. Congratulations! Your manuscript is now with our production department. 

Kind regards, 

on behalf of

Dr. Emily W. Harville 

Academic Editor

PLOS ONE